# rMELEISH: A Novel Recombinant Multiepitope-Based Protein Applied to the Serodiagnosis of Both Canine and Human Visceral Leishmaniasis

**DOI:** 10.3390/pathogens12020302

**Published:** 2023-02-11

**Authors:** Daniel Silva Dias, Juliana Martins Machado, Patrícia Aparecida Fernandes Ribeiro, Amanda Sanchez Machado, Fernanda Fonseca Ramos, Lais Moreira Nogueira, Ana Alice Maia Gonçalves, Luana de Sousa Ramos, Isadora Braga Gandra, Flaviane Silva Coutinho, Michelli dos Santos, Jonatas Oliveira da Silva, Miguel Angel Chávez-Fumagalli, Rafael Gonçalves Teixeira-Neto, Ana Thereza Chaves, Mariana Campos-da-Paz, Amanda A. Souza, Rodolfo Cordeiro Giunchetti, Sonia Maria Freitas, Sandra Lyon, Danielle Ferreira de Magalhães-Soares, Julia Angelica Gonçalves Silveira, Eduardo Sergio Silva, Eduardo Antonio Ferraz Coelho, Alexsandro Sobreira Galdino

**Affiliations:** 1Laboratório de Biotecnologia de Microrganismos, Universidade Federal de São João Del-Rei (UFSJ), Campus Centro Oeste, Divinópolis 35501-296, MG, Brazil; 2Programa de Pós-Graduação em Ciências da Saúde: Infectologia e Medicina Tropical, Faculdade de Medicina, Universidade Federal de Minas Gerais, Av. Prof. Alfredo Balena, 190, Belo Horizonte 30130-100, MG, Brazil; 3Laboratório de Biologia das Interações Celulares, Departamento de Morfologia, Instituto de Ciências Biológicas, Universidade Federal de Minas Gerais, Belo Horizonte 31270-901, MG, Brazil; 4Computational Biology and Chemistry Research Group, Vicerrectorado de Investigación, Universidad Católica de Santa María, Urb. San José S/N, Arequipa 04000, Peru; 5Laboratório de Doenças Infecto-Parasitárias, Universidade Federal de São João Del-Rei, Divinópolis 35501-296, MG, Brazil; 6Laboratório de Bioativos & Nanobiotecnologia, Universidade Federal de São João Del-Rei, Divinópolis 35501-296, MG, Brazil; 7Laboratório Nacional de Biociências (LNBio), Centro Nacional de Pesquisa em Energia e Materiais, Campinas 13083-970, SP, Brazil; 8Laboratorio de Biofísica, Instituto de Biologia, University of Brasilia, Brasília 70910-900, DF, Brazil; 9Fundação Hospitalar do Estado de Minas Gerais, Hospital Eduardo de Menezes, Belo Horizonte 30622-020, MG, Brazil; 10Departamento de Medicina Veterinária Preventiva, Escola de Veterinária, Universidade Federal de Minas Gerais, Belo Horizonte 31270-901, MG, Brazil

**Keywords:** leishmaniasis, recombinant chimeric protein, serodiagnosis, visceral leishmaniasis, humans, dogs

## Abstract

Background: visceral leishmaniasis (VL) is a critical public health problem in over ninety countries. The control measures adopted in Brazil have been insufficient when it comes to preventing the spread of this overlooked disease. In this context, a precise diagnosis of VL in dogs and humans could help to reduce the number of cases of this disease. Distinct studies for the diagnosis of VL have used single recombinant proteins in serological assays; however, the results have been variable, mainly in relation to the sensitivity of the antigens. In this context, the development of multiepitope-based proteins could be relevant to solving such problem. Methods: a chimeric protein (rMELEISH) was constructed based on amino acid sequences from kinesin 39 (k39), alpha-tubulin, and heat-shock proteins HSP70 and HSP 83.1, and tested in enzyme-linked immunosorbent (ELISA) for the detection of *L. infantum* infection using canine (*n* = 140) and human (*n* = 145) sera samples. Results: in the trials, rMELEISH was able to discriminate between VL cases and cross-reactive diseases and healthy samples, with sensitivity and specificity values of 100%, as compared to the use of a soluble Leishmania antigenic extract (SLA). Conclusions: the preliminary data suggest that rMELEISH has the potential to be tested in future studies against a larger serological panel and in field conditions for the diagnosis of canine and human VL.

## 1. Introduction

Leishmaniasis is a neglected disease caused by protozoan parasites of the genus Leishmania; the disease is endemic in 99 countries [1]. There are about 20 parasite species capable of causing the disease in humans, with an estimated annual incidence of 0.2 to 0.4 million cases of visceral leishmaniasis (VL) and 0.7 to 1.2 million cases of tegumentary leishmaniasis (TL) [2]. Over 90% of VL cases are registered in Bangladesh, Ethiopia, Brazil, India, Sudan, and South Sudan. TL is more widespread and occurs in countries such as Afghanistan, Algeria, Brazil, Iran, Peru, Ethiopia, North Sudan, Costa Rica, Colombia, and Syria [3]. In the Americas, Brazil accounts for more than 90% of documented VL cases, and the Leishmania infantum species is largely responsible for the disease in dogs and humans [2]. Human VL (HVL) and canine VL (CVL) represent serious public health risks because of asymptomatic infections in both hosts, and because domestic dogs are important reservoirs of the parasites and a source of infection to both vectors and humans [4,5]. 

The strategy adopted by the Brazilian Ministry of Health to control VL is based on early diagnosis, the treatment of human cases, the monitoring of seroreactive dogs, and vector control. However, such measures have been shown to have little effect [3,6,7]. The severity of the disease and the role of dogs as reservoirs emphasize the importance of monitoring and surveying *L. infantum* infections to prevent VL from spreading [8,9,10]. Thus, improving the diagnosis of both CVL and HVL is relevant to disease control and to formulating more effective public health policies.

A variety of laboratory methods have been used to diagnose VL in humans and dogs, such as indirect fluorescence antibody tests, enzyme-linked immunosorbent (ELISA), dot-ELISA, the direct agglutination test, Western blotting, and immunochromatographic assays [11,12,13]. There are currently six diagnostic HVL kits registered for sale in Brazil, most of which are imported. A recent study showed variable sensitivity and specificity in the performance of these tests, which was more pronounced in patients co-infected with Human Immunodeficiency Virus (HIV) [14]. Moreover, the cost-effectiveness of these tests varied significantly [15]. The Ministry of Health recommends CVL screening by the Dual-Path Platform (DPP; Bio-Manguinhos/Fiocruz, Rio de Janeiro, Brazil) and the confirmation of CVL cases using an ELISA kit (EIE-LVC kit; Bio-Manguinhos/Fiocruz, Rio de Janeiro, Brazil) [16,17]. However, the diagnosis of CVL cases may be underestimated, since the standard diagnosis for dogs in endemic areas presents low accuracy, considering that one in five seronegative dogs is infected [18]. Furthermore, when analyzing the available serological tests and quantitative polymerase chain reaction tests (qPCR), the level of agreement between the tests ranged from poor to moderate [19]. In this sense, the currently available diagnostic kits lack sufficient sensitivity and/or specificity, impairing the country’s health policy programs.

The ELISA is the candidate of choice for a rapid and reliable diagnostic test for Leishmania infection because it is practical, standardizable, and suitable for mass screening [20,21]. However, its specificity and sensitivity depend on the type and quality of antigen, and it could be improved by using recombinant multiepitope-based proteins since the variability in the humoral response found in humans and dogs is high. In this context, the combination of distinct antigens in a unique product could potentially improve diagnostic efficacy. Therefore, the selection of parasite antigenic proteins followed by the identification of B-cell epitopes and resulting in the construction of chimeric proteins could account for a more reliable diagnosis of CVL and HVL [5,22,23].

Such an experimental strategy has been shown to be a valuable basis for the serological diagnosis of diseases [5,23,24,25,26,27,28] and vaccines [29,30,31], where satisfactory results were obtained as compared to the use of soluble and/or crude antigenic extracts or isolated recombinant proteins. 

In this study, a recombinant multiepitope-based protein (called rMELEISH) was constructed based on amino acid sequences from kinesin 39 (k39), alpha-tubulin, and heat-shock proteins (HSP70 and HSP 83.1); the resultant chimeric protein was used for the detection of *L. infantum* infection in humans and dogs. The preliminary results described here showed that the recombinant protein was identified by antibodies in sera from canine and human VL, but not by samples from cross-reactive diseases or those from endemic controls.

## 2. Materials and Methods

### 2.1. Parasites

An *L. infantum* (MHOM/BR/1970/BH46) strain was used. Stationary promastigotes were grown at 24 °C in Schneider’s medium (Sigma, St. Louis, MO, USA), which was supplemented with 20% inactivated fetal bovine serum (FBS, Sigma, St. Louis, MO, USA), 20 mM L-glutamine, 200 U/mL penicillin, and 100 µg/mL streptomycin, at pH 7.4. The soluble Leishmania antigen extract (SLA) was prepared as described Coelho et al. 2003 [32]. Briefly, 10^9^ stationary-phase promastigotes per mL of *L. infantum* were washed 3 times in 5 mL of cold sterile phosphate-buffered saline (PBS). After 7 cycles of freezing (−196 °C) and thawing (+37 °C), the suspension was centrifuged at 8000× *g* for 20 min at 4 °C. The supernatant containing SLA was collected in 500 µL aliquots and then stored at −80 °C until use. The protein concentration was estimated using the Bradford method [33].

### 2.2. Canine Sera

The study was approved by the Ethical Committee on the Use of Animals in Research of the Federal University of Minas Gerais (UFMG), with the protocol number 244/2018. In total, 140 canine samples were used in this study. Blood samples were collected in vacutainer tubes without anticoagulant (BD Biosciences, Franklin Lakes, NJ, USA), and processed by centrifugation (5000× *g* for 10 min at room temperature); then, the sera were recovered from the supernatant, aliquoted, and stored at −70 °C, until use. Sera were collected from asymptomatic (*n* = 20) or symptomatic (*n* = 25) VL dogs, who were diagnosed by means of clinical evaluation and by a PCR technique using spleen and/or bone marrow aspirates to identify *L. infantum* kinetoplast DNA (kDNA). In addition, all VL animals presented positive serological results when evaluated using the EIE-CVL Bio Manguinhos^®^ kit (Rio de Janeiro, Rio de Janeiro, Brazil). Symptomatic animals presented three or more of the following clinical signs: emaciation, alopecia, anemia, conjunctivitis, dehydration, dermatitis, erosion, ulcerations, lymphadenopathy, and onychogryphosis. Otherwise, asymptomatic dogs were free of any sign of disease. Sera were also collected from healthy dogs living in endemic (*n* = 30, Belo Horizonte) and non-endemic (*n* = 20; Poços de Caldas, Minas Gerais, Brazil) areas of VL. These animals showed no clinical sign of leishmaniasis and presented negative serological results when evaluated with the EIE-CVL BioManguinhos^®^ kit (Rio de Janeiro, Rio de Janeiro, Brazil). Sera from Leish-Tec^®^-vaccinated dogs (*n* = 20) and from those experimentally infected with *Ehrlichia canis* (*n* = 15) or *Babesia canis* (*n* = 10), all of whom were kept in kennels to prevent their contact with transmitting vectors of leishmaniasis, were used.

### 2.3. Human Samples

The study was approved by the Human Research Ethics Committee of UFMG, with protocol number CAAE–32343114.9.0000.5149. In total, 145 human sera samples were used in the experiments. Blood samples were collected in vacutainer tubes without anticoagulant (BD Biosciences), and processed by centrifugation (5000× *g* for 10 min at room temperature); then, sera were recovered from the supernatant, aliquoted, and stored at −70 °C until use. Samples were collected from VL patients (*n* = 35), who were diagnosed by means of clinical evaluation and PCR to identify *L. infantum* kDNA in organic aspirates. In addition, all patients presented positive serological results when evaluated with the Kalazar Detect™ Rapid Test kit (InBios International Inc., Seattle, WA, USA). Sera were also obtained from healthy subjects living in areas where VL is endemic (*n* = 30); these subjects did not present any clinical sign of disease and showed negatve serological results. Samples from patients infected with Chagas Disease (*n* = 25), leprosy (*n* = 10), tuberculosis (*n* = 10), malaria (*n* = 10), and HIV (*n* = 25) were employed in the serological assays.

### 2.4. Design of the Synthetic Gene, Cloning, and Expression 

The amino acid sequence of the rMELEISH protein was submitted to the Protein Homology Recognition Engine V 2.0 server (Phyre2) [34] for modeling in an intensive mode. After modeling, the protein structure was visualized using the UCSF Chimera software (version 1.11.2) [35]. The synthetic gene was custom synthesized by Epoch Biosciences with codon usage for *E. coli*, and cloned as an *NdeI*/*XhoI* fragment into in-frame pET21a with a C-terminal histidine tag to allow for protein purification by affinity chromatography. The resulting plasmid was used to transform *E. coli* BL21 (DE3) *plysS* competent cells, and selection was performed on LB agar plates containing 100 µg/mL ampicillin. The DNA and amino acid sequences for the entire synthetic gene construct are proprietary (under Brazilian patent No. BR1020140313311) and cannot be shared at this stage. An individual colony was inoculated in 5 mL LB (10 g/L Casein Peptone, 5 g/L Yeast extract, 10 g/L NaCl, pH 7.2) containing 100 µg/mL ampicillin and allowed to grow overnight at 37 °C under agitation (200 rpm). Then, 1.25 mL of the pre-culture was transferred to 25 mL LB in a 250 mL Erlenmeyer flask. The culture was grown in the aforementioned conditions until an OD_600_ of 0.6, at which point 1 mM IPTG was added. Aliquots were withdrawn 0.5, 1.5, and 2.5 h after induction. The induced culture was harvested by centrifugation at 6000× *g* for 15 min at 4 °C and the pellet was stored at −80 °C. 

### 2.5. Purification of the rMELEISH Protein

The frozen pellet was resuspended in 1 mL lysis buffer (8 M urea, 50 mM NaH_2_PO_4_, 300 mM NaCl, 10 mM imidazole, pH 8.0) following incubation at 4 °C for 16 h. Cell suspension was then sonicated (5 pulses of 10 sec with 1 min intervals) using a Vibra Cell sonicator (Sonics & Materials, Inc, Newtown, CT, USA) and incubated on ice for 2 h following centrifugation at 6000× *g* for 15 min at 4 °C. The supernatant was added to 0.5 mL Ni-Sepharose 6 Fast Flow resin (Sigma,St. Louis, MO, USA) (resuspended in lysis buffer), which was then incubated at 4 °C for 90 min on a vertical disc rotator. Next, the resin was sedimented and washed 4 times with 1 mL washing buffer (8 M urea, 50 mM NaH_2_PO_4_, 300 mM NaCl, 5 mM imidazole, pH 8.0). The protein was eluted in 3 fractions using 0.5 mL elution buffer (8 M urea, 50 mM NaH_2_PO_4_, 300 mM NaCl, 50, 100, and 200 mM imidazole, pH 8.0).

### 2.6. Gel Electrophoresis and Western Blotting

Protein integrity and molecular mass calculation were evaluated by running sam-ples on 12% SDS-PAGE. Proteins were stained with Coomassie Brilliant Blue R-250 (Sigma-Aldrich). Following electrophoresis, the proteins were transferred electrophoretically to a Polyvinylidene difluoride (PVDF) membrane for Western blotting. The membrane was blocked with 5% skim milk powder in Tris-buffered saline with 0.1% Tween*^®^*20 (TBS-T), for 2 h at room temperature. It was then washed 3 times with TBS-T and incubated with monoclonal mouse anti-His AP (Alkaline Phosphatase, Sigma, St. Louis, MO, USA), diluted 1:1000 in TBS for 2 h at room temperature. Following 3 washes with TBS-T, the specific protein band was visualized using the nitroblue tetrazoli-um/5-bromo-4chloro-3′-indolylphosphate (NBT/BCIP) detection method.

### 2.7. ELISA Assay

Titration curves were plotted to determine the most appropriate antigen concentration and antibody dilution to be used. The wells of polystyrene plates (Sarsted) were sensitized with 35 ng purified rMELEISH protein, which was diluted in 100 µL 0.1 M sodium carbonate-bicarbonate buffer (pH 9.6). After incubation at 4 °C for 16 h, the coated wells were washed with PBST (PBS supplemented with 0.2% tween 20, pH 7.2) and blocked for 2 h at 37 °C with PBS containing 5% (*w*/*v*) dried skim milk powder and washed again with PBST. Subsequently, 100 µL of a dilution (100 µL PBST, 5% (*w*/*v*) dried skim milk powder, and 5 µL serum) was placed into the wells, resulting in a final dilution of approximately 1/20. After incubation for 1 h at 37 °C, the wells were washed with PBST and 100 µL of goat anti-dog IgG conjugate peroxidase-labeled diluted at 1:25,000 in PBS buffer (pH 7.2), containing 5% (*w*/*v*) dried skim milk powder, was added following incubation for 1 h at 37 °C. The wells were again washed with 100 µL TMB by incubating for 30 min at room temperature. The optical density (OD) values were read at 450 nm.

### 2.8. Fluorescence and Circular Dichroism Spectroscopy Assays

Conformational changes in the chimeric protein were evaluated by fluorescence spectroscopy using a Jasco FP-6500 Spectrofluorimeter (Jasco Analytical Instruments, Tokyo, Japan) coupled to a Jasco ETC-273T Peltier system (Jasco Analytical Instruments) with water circulation. The rMELEISH protein (0.05 mg/mL) was diluted in 10 mM sodium acetate buffer at pH 4.0 and 10 mM Tris-HCl, pH 7.0 and 9.0, at 25 °C. The excitation and emission slits were fitted at 5.0 and 10.0 nm, respectively. The excitation wavelength was 295 nm and emission spectra were recorded from 300–400 nm. The circular dichroism (CD) assays were performed using a Jasco J-815 spectropolarimeter (Jasco, Tokyo, Japan) equipped with a Peltier-type temperature controller and a thermostatized cuvette cell linked to a thermostatic bath. The far-ultraviolet (UV) CD spectra of the rMELEISH (0.1 mg/mL) in 2 mM acetate, pH 4.0, and 2 mM Tris HCl, pH 7.0 and 9.0, were recorded using a 0.1 cm-path-length quartz cuvette. After five consecutive measurements, the mean spectrum for each pH was recorded and the buffer contribution spectrum was subtracted. The ellipticities were converted into molar ellipticity (*θ*) based on the molecular mass of 115 Da per residue [36]. 

### 2.9. Statistical Analysis

Results were entered into Microsoft Excel (version 10.0) spreadsheets and analyzed using GraphPad Prism^TM^ (version 6.0 for Windows). Receiver–operator characteristic (ROC) curves were constructed using the OD values from VL sera versus those from negative or cross-reactive samples. The diagnostic performance of the antigens was evaluated by calculating the sensitivity (Se), specificity (Sp), area under the curve (AUC), and Youden index (*J*). Confidence intervals (CI) were defined using a 95% confidence level (95%CI). Differences were considered significant with *p* < 0.05.

## 3. Results

### 3.1. Characterization and Purification of the rMELEISH Protein

In order to design a multiepitope protein that could be of diagnostic use, linear and conserved B-cell epitopes, which were shown to be antigenic for VL and present specific antibodies against them, were selected, grouped in tandem, and connected by flexible glycine–serine linkers. This allowed the epitopes to be freely available for interaction with their cognate antibodies, thus contributing to the overall sensitivity and specificity of the diagnostic test. Some molecules were randomly repeated to increase epitope density, resulting in a chimeric protein called rMELEISH. After purification, the recombinant protein had a molecular weight of ~25.0 kDa (Figure 1C). A Western blotting experiment was conducted using an anti-histidine antibody to confirm the presence of the rMELEISH protein (Figure 1B). In addition, an SDS-PAGE 12% gel is shown (Figure 1A).

### 3.2. Tertiary and Secondary Structure of the rMELEISH Protein 

Fluorescence spectra of rMELEISH showed typical emission bands of solvent exposed tryptophan residues (Figure 2A). The intensity of the emission bands was similar in all pH ranges; however, a red shift of 2 nm (340 to 342 nm) was observed at pH values of 4.0 and 9.0, relative to the emission band at pH 7.0. Far-UV CD spectra of rMELEISH exhibit a dichroic signal close to 0 at pH 4.0 and 9.0 (Figure 2B), without characteristic secondary structure bands, and a very low signal appears at pH 7.0.

### 3.3. Diagnostic Evaluation of rMELEISH for Human VL

Initially, the diagnostic efficacy of the rMELEISH protein was preliminarily evaluated using ELISA experiments for human VL. *L. infantum* SLA was used as comparative antigen. The results showed that the chimeric protein was recognized by antibodies in VL patients’ sera, presenting a low reaction in healthy subjects and in cross-reactive patient sera (Figure 3). ROC curves were constructed with the individual OD values for each antigen, and the results showed 100% sensitivity and specificity for rMELEISH, while the values for SLA were 91.4 and 76.4%, respectively (Table 1). 

The preliminary diagnostic efficacy of rMELEISH was then evaluated for canine VL. The results showed that the chimera was recognized by all asymptomatic and symptomatic VL dogs’ sera, but not by healthy dogs or cross-reactive sera (Figure 4). Otherwise, SLA was poorly recognized by asymptomatic VL sera, and presented high cross-reactivity. ROC curves were constructed and the results for rMELEISH showed sensitivity and specificity values of 100 and 100%, respectively, and 51.11 and 88.42%, respectively, for SLA (Table 1).

## 4. Discussion

Visceral leishmaniasis is a major public health problem that can be fatal if acute and left untreated [37]. In Brazil, the disease is an endemic zoonosis and, although the Ministry of Health has proposed several control measures to prevent its spread, they present variable efficacy. As a result, the number of cases in dogs and humans has increased in recent years [38]. Accurate early diagnosis is necessary for the more adequate management and control of VL, as a late diagnosis is associated with disease progression and death [39,40]. However, the available diagnostic techniques present variable sensitive and/or specificity, mainly due to such factors as the degree of infection, the type and quality of antigen, and the populations evaluated [18].

Among the serological tests employed for the VL diagnosis, ELISA is commonly used for the detection of symptomatic cases; however, its sensitivity and specificity vary when it comes to detecting asymptomatic cases. In addition, crude or soluble promastigote and amastigote antigens present cross-reactions, which limit their utility. Single proteins applied in a recombinant format have improved the specificity of the tests, although the sensitivity remains variable. A promising solution to this problem is based on the use of multiepitope-based proteins, which could maximize sensitivity and specificity in a single product, thus reducing production costs [41]. The construction of such recombinant proteins containing high density epitopes represents an alternative method for the diagnosis of diseases, given that these proteins exhibit a broad ability to expose their epitopes more efficiently, resulting in improved sensitivity and specificity [42]. 

In this work, a chimeric protein was developed and evaluated as a strategy for the diagnosis of VL in dogs and humans. The study focused on three evolutionarily conserved protein families to construct the gene-encoding rMELEISH protein: kinesins, heat shock proteins, and tubulins. Alpha-tubulin is well-known for its participation in the immune response of mammals [43]. The K39 antigen is a 39-amino-acid-repetitive immunodominant protein that has also been also shown to present antigenicity in mammals, and which is capable of detecting human VL and symptomatic and asymptomatic cases of canine VL [44,45,46,47,48,49,50]. Meanwhile, heat-shock proteins, such as HSP70 and HSP83.1, have also been shown to be involved in the humoral response of infected patients [51,52,53,54,55,56,57]. Such antigens were evaluated using information available in the literature, with the main B-cell epitopes being selected based on the following criteria: they had to be (i) immunodominant, (ii) specific for anti-VL antibodies, and (iii) phylogenetically conserved in distinct *Leishmania* species. 

A few studies in the literature use recombinant multiepitope-based proteins to diagnose canine and human VL. A search of Pubmed conducted using the keywords: “Multiepitope protein” and “diagnosis” identified 268 published articles. Out of these, only three were related to the diagnosis of canine and human VL [5,23,25]. Faria et al. 2015 [5] described the use of two multiepitope-based proteins, PQ10 and PQ20, and their diagnostic potential to detect asymptomatic dogs (80%) infected with *L. infantum*; they were more effective than the EIE-LVC Kit. The ELISA sensitivity of the PQ10 and PQ20 proteins was 88.8% and 84.9%, respectively, with specificities of 80% and 65%, respectively. Dalimi et al. 2020 [25] reported a multiepitope-based protein, PQ10, which was able to distinguish asymptomatic from symptomatic dogs infected with *L. infantum* from other groups, with sensitivity and specificity values of 94% and 86%, respectively. Yaghoubi et al. 2021 [23] studied the use of a multiepitope-based protein, P1P2P3, for the diagnosis of VL. The authors demonstrated that the protein presented 98% and 95.3% sensitivity and specificity, respectively. The use of recombinant proteins can reduce the number of false-negative reactions. Vale et al. 2021 [58] evaluated the performance of a recombinant chimera compared with two commercial kits for the diagnosis of canine visceral leishmaniasis. False-negative reactions were not observed using the evaluated recombinant antigen; however, in the commercial kits, 40.0 and 16.0% of results were false negatives. No cross-reactivity was found in this study with healthy samples or dogs and humans. In addition, the protein used in this study was able to identify both asymptomatic and symptomatic dogs, with 100% values, as well as diagnosing the disease in humans with the same value. 

The fluorescence spectra of the rMELEISH in all analyzed pH values were typical of tryptophan completely exposed to a polar solvent [59]. The similar intensity and the 2 nm red shift of the emission bands indicated very small conformational changes in the protein as a function of pH. In addition, the far-UV CD spectra demonstrated a typical unstructured protein at pH values of 4.0 and 9.0, and a very low profile of structured protein at pH 7.0. These results are in agreement with the recombinant multiepitope protein that was constructed to form an unstructured molecule for hepatitis B diagnosis [60].

The limitations of the present study include the absence of a comparative ELISA assay with commercial antigens or diagnostic kits, as well as the small sample sizes for both canine and human sera. Therefore, the results presented here could be considered as a proof-of-concept of the use of this recombinant chimeric protein as a diagnostic antigen for VL, although additional works are certainly necessary in order to prove the efficacy of this antigen in identifying *Leishmania infantum*-infected dogs and humans.

## 5. Patents

One patent resulting from the work reported in this manuscript is under protection at Instituto Nacional de Propriedade Industial (INPI): Brazilian patent No. BR1020140313311.

## Figures and Tables

**Figure 1 pathogens-12-00302-f001:**
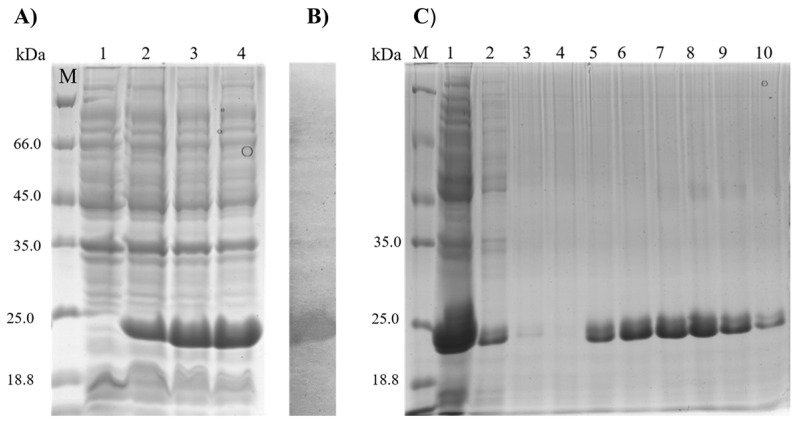
SDS–PAGE analysis of the rMELEISH. (**A**) 12% SDS–PAGE of the accumulation of the rMELEISH post induction with 1 mM IPTG. Lane M: Unstained Protein Weight Marker Standards (Thermo Scientific, Waltham, MA, USA) Lanes 1–4; induction times of 0, 0.5, 1.5, and 2.5 (h), respectively. (**B**) Western blotting analysis of the crude extract after 2.5 h of induction. (**C**) The protein was eluted in 5 fractions using 0.5 mL elution buffer containing different concentrations of imidazole. M: Unstained Protein Weight Marker Standards (Thermo Scientific, Waltham, MA, USA), Lane 1: flow through (FT); lanes 2–4: wash; lanes 5–6: elutions with 50 mM imidazole; lanes 7–8: elutions with 100 mM imidazole, which is considered the best purification condition, and lanes 9–10: elutions with 200 mM imidazole.

**Figure 2 pathogens-12-00302-f002:**
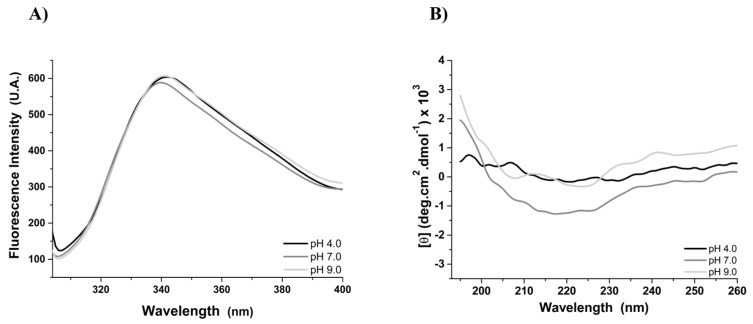
Structural analysis of rMELEISH by fluorescence and circular dichroism. (**A**) Fluorescence spectra of the rMELEISH as a function of pH 4.0, 7.0, and 9.0 at 25 °C. The fluorescence emission bands were red shifted by 2 nm from 340 nm (pH 7.0) to 342 nm (pH 4.0 and 9.0). (**B**) Far-UV CD spectra of the rMELEISH at pH 4.0, 7.0, and 9.0 and 25 °C. The typical CD bands of the secondary structure were not observed at pH values of 4.0 and 9.0 and a very low signal appears at pH 7.0.

**Figure 3 pathogens-12-00302-f003:**
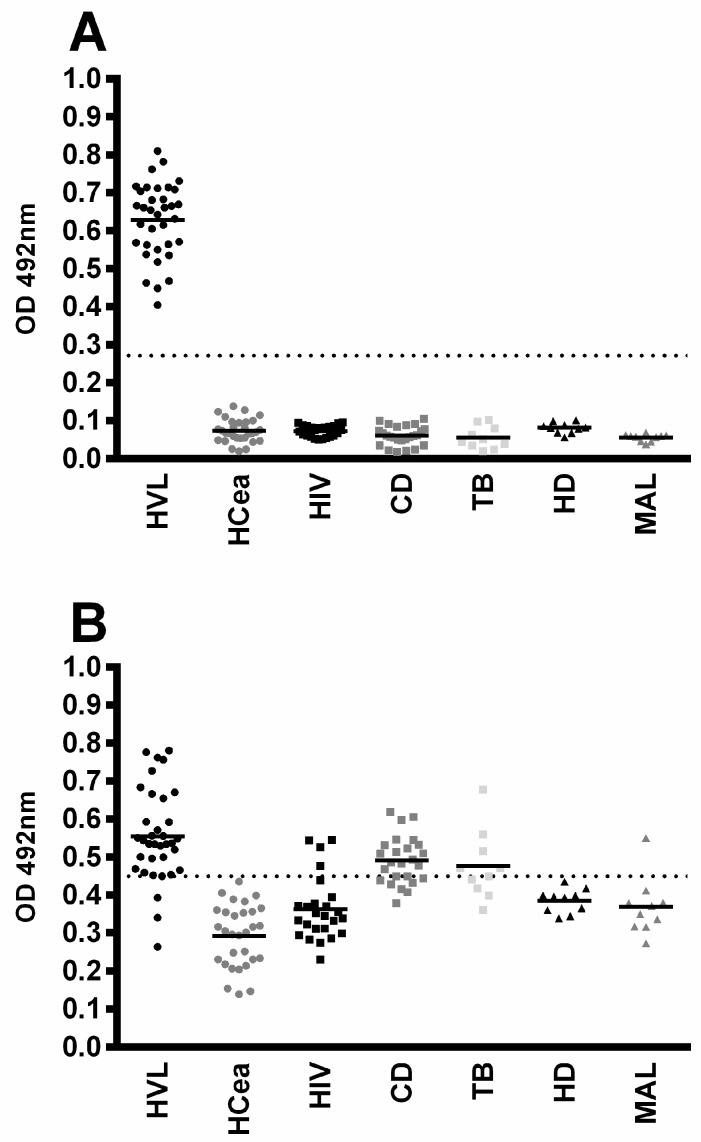
ELISA reactivity for the diagnosis of human visceral leishmaniasis. ELISA assays were performed using sera samples from visceral leishmaniasis patients (HVL; *n* = 35), sera from healthy subjects living in regions where the disease is endemic (HCea; *n* = 30); and sera from patients with Chagas Disease (CD, *n* = 25), leprosy (HD, *n* = 10), tuberculosis (TB, *n* = 10), malaria (MAL, *n* = 10), and HIV (HIV; *n* = 25). ROC curves were constructed with the individual OD values for each serum sample against rMELEISH (panel **A**) or *L. infantum* SLA (panel **B**), and the data are shown. The dotted lines represent the cut-off value obtained by the ROC curves, which were used to obtain the sensitivity, specificity, and AUC of the antigens. The mean of each group is also shown.

**Figure 4 pathogens-12-00302-f004:**
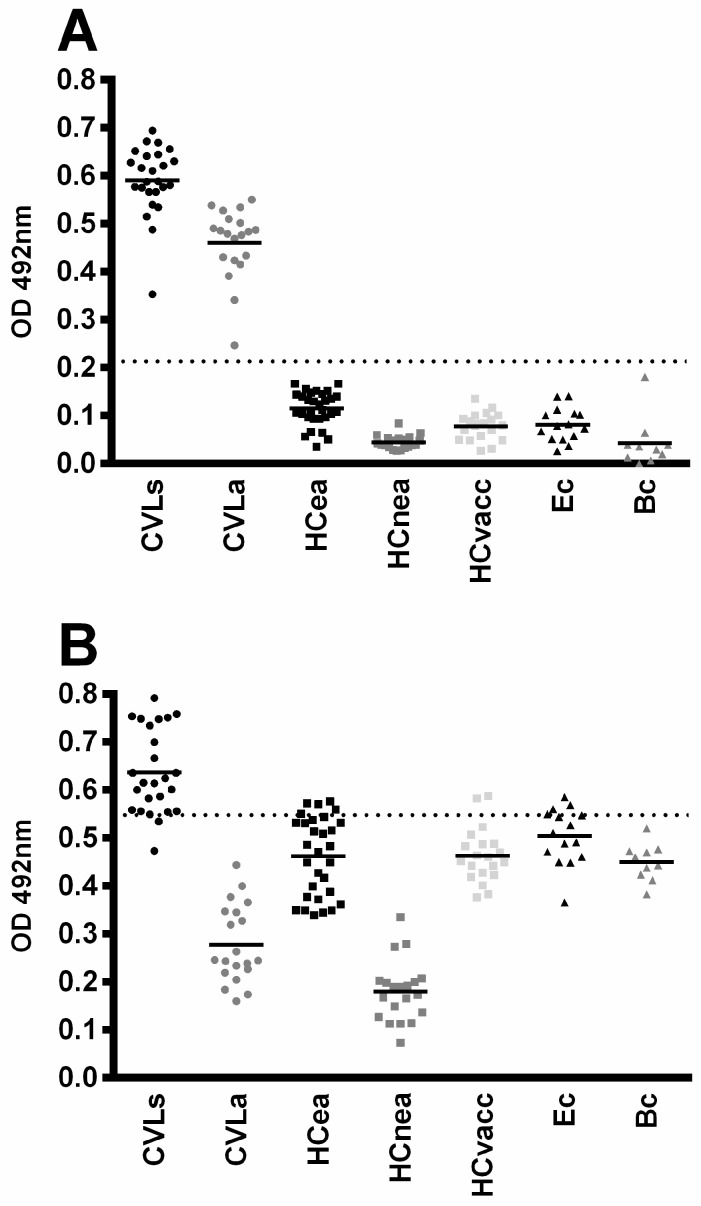
ELISA assays using the antigens for the diagnosis of canine visceral leishmaniasis. ELISA experiments were performed using sera from asymptomatic (CVLa; *n* = 20) and symptomatic (CVLs; *n* = 25) visceral leishmaniasis dogs, sera from healthy dogs living in endemic (HCea; *n* = 30) or non-endemic (HCnea; *n* = 20) areas, or those immunized with the Leish-Tec^®^ vaccine (*n* = 20); samples were additionally taken from animals infected with *Ehrlichia canis* (Ec) *n* = 15) or *Babesia canis* (Bc) *n* = 10). ROC curves were constructed with the individual OD values for each serum sample against rMELEISH (panel **A**) or *L. infantum* SLA (panel **B**), and the data are shown. The dotted lines represent the cut-off value obtained by the ROC curves, which were used to obtain the sensitivity, specificity, and AUC of the antigens. The mean of each group is also shown.

**Table 1 pathogens-12-00302-t001:** Diagnostic evaluation of the antigens for canine and human visceral leishmaniasis. Sera samples were used in ELISA experiments against rMELEISH and *L. infantum* SLAs, in order to obtain the individual optical density values. ROC curves were constructed and the diagnostic efficacy was evaluated by calculating the sensitivity (95% CI), specificity (95% CI), area under the curve (AUC), and Youden index (J).

	Canine Sera							
Antigen	AUC	*p*-Value	Cut-Off	Se	95%CI	Sp	95%CI	J
rMELEISH	1.0	<0.0001	>0.2130	100	92.13–100	100	96.19–100	1.0
SLA	0.63	0.017	>0.5480	51.11	35.77–66.30	88.42	80.23–94.08	0.39
	**Human Sera**							
**Antigen**	**AUC**	***p*-Value**	**Cut-Off**	**Se**	**95%CI**	**Sp**	**95%CI**	**J**
rMELEISH	1.0	<0.0001	>0.2712	100	90.00–100	100	96.70–100	1.0
SLA	0.86	<0.0001	>0.4493	91.43	76.94–98.20	76.36	67.32–83.94	0.68

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
