# Peer review of "rMELEISH: A Novel Recombinant Multiepitope-Based Protein Applied to the Serodiagnosis of Both Canine and Human Visceral Leishmaniasis"

_pathogens, 2023, doi:10.3390/pathogens12020302_

Round 1

Reviewer 1 Report

The authors present a recombinant protein consisting of several epitopes as part of the diagnosis of HVL and CVL.

Scientific names should be written in italics.

Line 87: ELISA (enzyme-linked immunosorbent assay), therefore, it is not necessary to repeat the word "assay".

Lines 153-155 indicate the number of a patent. Its necessary to check if according to the legislation it is possible to publish the characteristics of the antigens generated. 

Figure 1 shows the purification of the protein obtained. However, in panel C additional components to the main band are observed. Additional purity analysis is not mentionated.

Important references related to the work are missing, such as Serodiagnosis of canine leishmaniasis using a novel recombinant chimeric protein constructed with distinct B-cell epitopes from antigenic Leishmania infantum proteins, Veterinary Parasitology (2021) https://doi.org/10.1016/j.vetpar.2021.109513.

The universe of sera tested is small, It is necessary to include a larger number of samples, as well as controls to determine the robustness of this assay.

Author Response

Dear Editor and Ms. Željka Nikolić,

As agreed in my email from january 12th, I uploaded the modified version today on behalf of all authors.

We were able to address all points raised and, thus, hope that our responses will satisfy your editorial board.   Please let me know if you have any further questions. We look forward to hearing from you.   My very best, Prof. Alexsandro Galdino

Reviewer 2 Report

Good manuscript

Recommend publication with minor corrections

1.     Line 114- make it superscript ---109

2.     Line 118- SLA was collected in 500 mL aliquots. Is this volume correct?

3.     138- why was heat inactivation (complement inactivation) was done for human serum samples? In real-time HI is typically not carried out.

4.     Also HI done only for human and not for canine samples. This, makes a discrepancy. Any explanation for points 3 & 4?

5.     140- clinical evaluation and PCR to identify L. infantum. Are all human samples (n=35) tested positive with PCR? If not clinical evaluation only will not confirm as a positive case. Why were they not tested for seropositivity? This will add your confirmation as true positives. Please clarify

6.     141 Line -Endemic negative control were subjected to only sero diagnosis? Why were they not subjected to PCR as your VL positive group? Please clarify. Your positive and negative sample identification is not consistent with both human and canine groups. This has to be corrected.

7.     Although abstract& discussion mention about rMELEISH k39, alpha-tubulin HSP 70 & HSP 83.1 AA sequence, nothing is mentioned in methods. Please include.

8.     Line 300- VL at start should be expanded

Author Response

(The authors gave the same response as above.)
